# Method of Building RAG-Powered Instruction Dataset from Raw Corporate Text Data for LLM Fine-Tuning

**Vadim Eliseev & Aleksandra Maksimova**
Laboratory of Intelligent Systems
Institute of Applied Mathematics and Mechanics
Donetsk, Russian Federation
`{eliseevv02,maximova.alexandra}@mail.ru`

**Vitaly Bondarenko**
Physics and Technical Faculty
Donetsk State University
Donetsk, Russian Federation
`mail@vitbond.ru`

## Abstract

The emergence of powerful LLMs enables the creation of high-quality intelligent assistants for QA search over corporate data. There are two commonly used approaches for adapting LLMs to domain-specific data: fine-tuning and Retrieval Augmented Generation (RAG). A RAG system involves building a vector database from corporate data split into chunks of a certain size using retriever. The system then finds the most relevant chunks according to user query and passes them to a generator to extract the answer. Separately, neither method guarantees the highest quality of the final response. LLM fine-tuning suffers from the need for frequent retraining to update its knowledge, which is a costly and complex process. Also, naive RAG suffers from limitations inherent to retriever work. In this paper we present the method of building instruction dataset for fine-tuning generator LLM as a part of RAG system in corporate intelligent assistant. As an example, the instruction dataset is generated from raw corporate data of Donetsk State University (DonSU) in Russian. We suggest combining the naive RAG approach with fine-tuning an LLM on a specific instruction dataset. A third component, a relevant context retrieved from the vector database, is added to the standard input-output instruction pairs. Proposed method involves five steps: 1) collecting raw text data from university website; 2) selecting relevant texts; 3) generating synthetic input-output instruction pairs; 4) building a vector database from the relevant texts; 5) supplementing the generated pairs with context retrieved from the vector database based on similarity to the input. Fine-tuning the LLM on such a dataset will not only make it to memorize domain-specific data, but also will improve its ability to interpret information, provided by the retriever from the vector database. This is a key aspect of the generator in the RAG system, and this training approach will ensure more accurate performance of the assistant. As a result, we created the RAG-powered instruction dataset from raw corporate text data about DonSU. It contains over 25,000 input-context-output records, specifically prepared for training an LLM that will serve as the generator in the developed intelligent assistant.

## 1 Introduction

Timely provision of up-to-date information is crucial in the educational process at universities. This is especially important during the admissions period when a large number of applicants request similar reference information. With the emergence of language models (LLMs) such as GPT-4 Achiam et al. (2023), which is accessible via API[1], LLaMA Touvron et al. (2023), an open-source model[2], etc., it has become possible to automate this communication process by creating an intelligent assistant capable of conducting a question-answer (QA) search over corporate data. This concept is

---

[1] https://platform.openai.com
[2] https://huggingface.co/meta-llama

not limited to the university domain and can be implemented in any field where QA search over a knowledge base is applicable.

Despite the great capabilities of modern LLMs, they usually do not have any knowledge of internal processes in most organizations unless this information is publicly available and included in the model's training data. This makes incorporating such "hidden" knowledge into LLMs a challenging task for developers of intelligent assistants.

There are two main approaches to incorporating domain-specific knowledge into LLMs: fine-tuning them Gururangan et al. (2020) or creating Retrieval Augmented Generation Systems (RAG) Lewis et al. (2020). These approaches enable the chosen LLM to work with domain-specific data that was not encountered during pretraining, with a certain degree of effectiveness. However, both of them suffering from their key drawbacks, that makes their separate use less efficient Zhang et al. (2024). These drawbacks will be described in more detail in Section 2.

In this paper we describe an approach that combines supervised fine-tuning and RAG. Additionally, we present a method of building a dataset which enables an LLM tuned on it to effectively function as a generator within a RAG system. Instead of using a classical instruction datasets consisting of input-output pairs, where the input represents a potential user question or instruction and the output is the model's target response, we propose using a dataset composed of input-*context*-output records. In this setup, the *context* serves as external information from which the generator model must extract the most relevant information and generate its response.

Our experiments are based on data from Donetsk State University (DonSU) website[3] and involve five steps:

1. Collecting text data from the website;
2. Selecting relevant text files to proceed;
3. Generating synthetic instructions using an LLM;
4. Building a vector database from the extracted text files;
5. Adding context according to the input from the built vector database to generated instructions as the *context* field.

The source code of each step and experiment is open-source, links will be provided in the corresponding sections.

## 2   RELATED WORK

The most common approaches for adapting LLMs to domain-specific data are either Retrieval Augmented Generation (RAG), which serves as the simplest baseline to implement, or supervised fine-tuning, which is often a complex and costly process.

The first mention of RAG was introduced by the FAIR team in 2020 Lewis et al. (2020), where authors proposed using two models for question answering: a retriever (specifically, Dense Passage Retriever) and a generator — a transformer model, with the authors suggesting BART-large Lewis (2019), a pre-trained seq2seq transformer Vaswani (2017) with 400M parameters.

The role of the retriever is to vectorize prepared text chunks containing domain-specific information. Text vectorization allows the construction of an n-dimensional representation that captures the semantic essence of the text. The first studies on the properties of text vector representations were presented in Mikolov et al. (2013) during the building of word2vec language model. The retriever also encodes the user query in a way that document chunks most likely containing the answer have the highest vector similarity to the query. Then the user query, along with the most relevant context chunks, is passed to the generator model which generates human-readable text that serves as the response.

This approach was later referred to as "Naive RAG" and was subsequently improved by introducing pre- and post-retrieval processing, adding metadata to text chunks, and ranking retrieved chunks

---

[3]https://donnu.ru

Gao et al. (2023) before passing to generator model. One of the main limitations of RAG is the imperfect performance of the retriever Barnett et al. (2024), which can cause the situations when the generator model is unable to extract a relevant response from the retrieved document chunks. This may happen if none of the chunks contain the answer or if they include information about different concepts weakly correlated with the query's semantics, causing the model to generate incorrect information. The best way to deal with such situations is for the model to acknowledge that it lacks the information needed to generate the answer.

The other popular approach for adapting LLMs to domain-specific data is fine-tuning. Previously, assistant models were fine-tuned using dialogues either manually annotated by people or extracted from external sources, such as Reddit comments and Twitter conversations Zhang (2019). When the instruction-based approach of LLM training emerged, researchers from OpenAI fine-tuned GPT-3 on manually annotated instructions and introduced InstructGPT Ouyang et al. (2022), which became the predecessor of ChatGPT. However, collecting a manually annotated instruction dataset for training assistant models in non-industrial settings is hard and costly. Therefore, synthetic datasets generated by powerful LLMs, or obtained using the Self-Instruct approach Wang et al. (2022), are commonly used for this purpose.

There are several techniques for fine-tuning LLMs, such as fine-tuning all model weights, which can be costly and infeasible in non-industrial settings; adapting specific model layers while freezing the rest Li & Liang (2021); training the bias vectors while freezing everything else Zaken et al. (2021); adapter tuning Houlsby et al. (2019); and others. The best fine-tuning approach, given the available resources for implementing an assistant, is Low-Rank Adaptation (LoRA). LoRA enables high-quality results by training a relatively small number of parameters while also providing flexibility in modifying the model's behavior during runtime Hu et al. (2021).

However, when we tune an LLM on a domain-specific dataset containing standard instruction-based input-output pairs, we make the model simply memorizes the data. This is not an ideal solution for building a QA system for a university, as university-related information frequently changes. As a result, we would either need to continuously retrain the model on up-to-date data, which is costly and complex, or rely on lately added external sources, which the model, after instruction tuning, would struggle to utilize effectively.

## 3  METHOD

To overcome the limitations, that can be caused by separate usage of RAG or fine-tuning, we propose combining these approaches. We suggest training the generator model not only to answer questions based on instructional input-output pairs from domain-specific data, but to extract the most relevant information from text chunks provided by the retriever model, according to the user's query. This method of fine-tuning has been compared with preparing to the exam using both the "open book", which refers to external knowledge database, and a basic student's memory, which refers to inductive biases of the model Zhang et al. (2024).

To the standard instructional input-output pairs we add a third component - relevant context retrieved from the vector database built on raw corporate text data. Passing this data to the generator model during fine-tuning will enhance its extracting and generating capabilities. Moreover, the generator's responses will become more domain-specific due to the tuning of the model on corporate data. Additionally, we will be able to update our vector database freely, and the generator model, thanks to RAG-powered tuning, will generate responses that closely match the quality of training outputs, since generalization will not rely on memorizing factual knowledge but rather on extracting relevant information from the provided text chunks in accordance with the specifics of the target domain.

This idea has been analyzed by Liu et al. (2024). Their study was conducted to train proprietary LLMs to function as generators in RAG systems using open-text corpora of *various* content in English with a dataset in format similar to ours. The other research Zhang et al. (2024) also was focused on tuning the LLM for RAG application using the vector context on open datasets from various domains. The authors of RAFT also proposed adding the negative documents (documents with low similarity with the query) in order to learn model to exclude the irrelevant documents from the final answer. The authors of Lin et al. (2024) separately tuned retriever and generator models also in order to achieve better performance of the RAG system over the open multi-domain data.

In our research, we focused on constructing a dataset from scratch using the raw single *domain-specific* text data about DonSU in Russian, using the university's website as the source of data.

We separately collected textual data from HTML pages of the university's website and text documents in various formats (such as .pdf, .docx, .pptx, etc.). The data from HTML files was used for two main purposes. First, it was used to generate instructional pairs, which formed the standard dataset for model fine-tuning. Second, it was used to build the vector knowledge database, which served as the source of relevant context chunks retrieved for each input from the generated instructional pairs. The collected text documents will be added to the vector database before running inference in the RAG system. As a result, our model will be fine-tuned on a relatively small portion of corporate data. This will allow us to evaluate the effectiveness of our approach when scaling the vector database to the entire available text corpus.

In this paper we focus on the process of building the RAG-powered instructional dataset and describe each required step in this process. We leave the actual fine-tuning of the generator model and its testing within the RAG system for future research. To build the RAG-powered dataset in the described format, the following steps were performed:

1. Crawled textual data from the university's website;
2. Selected the relevant text files that can be used for generating of instructional pairs and to build the vector context database;
3. Generated the synthetic instructional pairs from the relevant text files selected in the previous step using an LLM;
4. Built a vector database from the relevant text files selected in the previous step;
5. Added context from the built vector database to the generated instructional pairs based on the input field, and labeled it as the *context* field.

All steps, along with links to the source code of their implementation, will be described in the following sections.

## 4   COLLECTING OF RAW TEXTUAL DATA

In this section, we describe the process of collecting raw textual data from the DonSU website, as well as the process of selecting relevant text files suitable for building the instructional pairs and vector context dataset.

### 4.1   CRAWLING OF DATA FROM UNIVERSITY'S WEBSITE

Using the university's website as the source of domain-specific data for building an intelligent assistant is useful because it is a single official source containing structured and valid information. Also, according to local regulations by for Supervision in Education & Science (2024), the content of university websites in Russia is strictly regulated both in terms of the required sections and the acceptable file formats. This makes it possible to use a single parsing algorithm for websites of different universities.

We developed a set of scripts that is available at `https://github.com/EliseevVadim/WebsiteCrawler`. They allow both parsing data from a specified website and performing basic preprocessing of the generated text files, such as removing empty ones and merging all text files into a single document, which is useful for model fine-tuning scenarios based on next-token prediction Brown et al. (2020). The script accepts a content selector as one of its arguments, which determines the block where the textual content is located on the website's pages. This standardization is also dictated by regulations, making university websites machine-readable. Additionally, the user can specify the website traversal depth; if not set, the script performs a full crawl of all pages or stops after reaching the specified depth. Moreover, it is possible to specify whether textual files in various formats should be downloaded instead in addition to simply saving HTML pages as text.

After running, the script starts processing from the website's main page and recursively follows all discovered links. If a URL points to an HTML page, the script extracts its textual content and saves it to a file. If the URL leads to a downloadable file, it is fetched, provided that the user has enabled

this option. While parsing HTML pages, the script also saves their titles and last update dates. These metadata can be useful for both generating instructions and ranking retrieval results from the vector database.

As a result of crawling the website on December 11, 2024, a total of 2,912 text files were generated based on the extracted content of parsed HTML pages. Additionally, 3,546 files of various formats were downloaded, with a total data volume of 14.3 GB, which is sufficient for building a knowledge base to support the intelligent assistant.

## 4.2 SELECTION OF RELEVANT TEXT FILES

Initial analysis of the generated text files showed that some of these files do not contain any relevant information suitable for either generating instructional pairs or using them inside the vector database during RAG system inference. Most of these files originated from intermediary pages that contain only links to other pages on the website or direct links to downloadable files. For further research, such files need to be excluded.

We designed a prompt where it is possible to insert the content of a specific text file generated in the previous step and its title, which corresponds to the title of the page that originated this file. The prompt requests an external LLM to evaluate the provided text in terms of its usefulness for building an instruction dataset intended for fine-tuning an LLM. Additionally, the prompt includes a rating scale that allows each file to be marked accordingly. This scale is illustrated in Figure 1.

**1: Текст не имеет смысловой нагрузки, состоит только из ссылок или является переходником без полезной информации.**

**2: Текст содержит ссылки и заголовки с минимумом релевантной информации, включая старые или неактуальные данные.**

**3: Текст пригоден для создания ограниченного числа (менее 5) общих инструкций, но с низкой ценностью.**

**4: Текст подходит для создания значимого числа инструкций, отражающих как общие, так и частные аспекты работы организации, с достаточной долей актуальных данных.**

**5: Текст высоко структурирован, содержит множество актуальных данных (с 2022 года и новее), позволяет создавать множество разнообразных и ценных инструкций.**

Figure 1: Estimation scale used in prompt for relevant texts selection.

The full prompt in Russian can be found at: `https://github.com/EliseevVadim/TextsEstimator/blob/main/prompts/texts_evaluation.txt` The English translation of the prompt is provided in Appendix A.

Next we selected three LLMs: Gemma2-9b-it Team et al. (2024), LLaMA-3.1-70B-Instruct[4] and LLaMA-3.3-70B-Instruct[5], and sent prompts for all available text files to each model using the NVIDIA LLM inference service[6] via the OpenAI API. Additionally, we aimed to determine which of the available LLMs would be the most suitable for generating instructions. Since some models had a limit on the input sequence size in tokens, we decided to evaluate only files smaller than 11KB (this threshold was chosen based on an analysis of file size distribution). For all larger files, we assigned a score of **5.0**, as intuitively, the larger the file, the higher the likelihood that it contains useful content.

After running the script, we obtained distributions of file evaluations, shown in Figure 2. The figure consists of three histograms labeled (a), (b), and (c), each representing the evaluation scores assigned by different models.

---

[4]https://huggingface.co/meta-llama/Llama-3.1-70B-Instruct

[5]https://huggingface.co/meta-llama/Llama-3.3-70B-Instruct

[6]https://build.nvidia.com/

**Distribution of file evaluations**

Figure 2: Distribution of file evaluations from each model.

As we observe, all score distributions are skewed to the right, with the mode corresponding to the value just before the maximum. Interestingly, the Gemma2-9b-it never assigned a score of **5.0**, which may indicate its weaker suitability for this type of text analysis. The scores from the LLaMA family models are mostly similar. The latest model, LLaMA-3.3-70B-Instruct, predominantly assigned scores of **4.0**.

Next, we calculated the average scores assigned by the models for each text file and determined the maximum difference in scores given to the same file by different models. The highest "conflict" between model scores was **3** points, which was observed in 14 files. However, after manually reviewing these cases, we decided not to exclude them from the dataset used for instruction generation and vector database construction.

After analyzing the distribution of average file scores, we decided to discard all files with an average score of $\leq 2.0$. A score at this level indicates that either **all models assigned a 2.0**, suggesting the file is nearly unusable for further processing, or **most models rated it 1.0**, marking it as completely irrelevant. After removing these non-relevant text files, 2,337 files remained, which is sufficient for building the instruction dataset. Assuming an average of 10 instructions per file, this results in a dataset of over 23,000 instruction pairs. This is a strong result, considering our domain is primarily limited to data about a specific university. For comparison, the authors of Liu et al. (2024) constructed a dataset of 40,000 entries using data from the entire Wikipedia.

We also tested our hypothesis about the relationship between file size and average score by plotting a correlation matrix, shown in Figure 3.

As seen, the correlation between the average score and file size is positive and amounts to **66%**, indicating a relationship between these metrics and justifying the assignment of the highest score to "large files". It is also noticeable that the LLaMA-class models had the greatest impact on the average score, suggesting their greater suitability for analyzing *crawled* raw text data. Based on this, we will use the LLaMA-3.3-70B-Instruct for instructional pair generation. The source code of the project containing files evaluations is available at: `https://github.com/EliseevVadim/TextsEstimator`

## 5 GENERATING INSTRUCTIONS

After selecting relevant files, we can now create a classical synthetic instructional dataset using LLM prompting. Unlike the previous section, we will use only one model — LLaMA-3.3-70B-Instruct, as it demonstrated the best performance in evaluating text files.

We designed a prompt that allows us to insert the file's title, its content, and the last update date. The model is tasked with generating the maximum possible number of domain-specific instructional pairs while considering suggested thematic directions and constraints. A fragment of the prompt is shown in Figure 4.

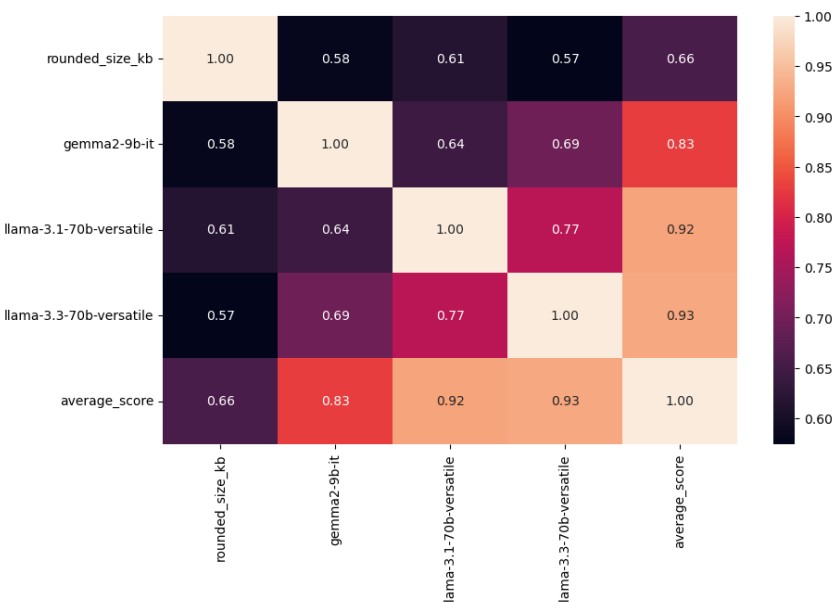

Figure 3: Correlation matrix between file size and model scores.

**Возможные типы инструкций:**

**1. Вопросы на поиск фактов или деталей из текста (при наличии фактологического материала вопросами необходимо покрыть их все).**

**2. Вопросы, требующие анализа или сравнения информации.**

**3. Любые другие вопросы, применимые для дообучения интеллектуального ассистента ДонГУ.**

**Допускается вывод только в формате JSON, любое другое текстовое содержимое строго запрещено! Вывод должен осуществляться исключительно на русском языке! Все ФИО упомянутых людей, а также любые названия, представленные на английском/других языках (кроме контекстуально непереводимых) должны быть обязательно переведены на русский язык!**

Figure 4: Part of the prompt for generating instructional pairs from relevant text files.

The full prompt in Russian can be found at: `https://github.com/EliseevVadim/InstructionsGenerator/blob/main/data/prompts/instructions_creation_RU.txt` The English translation of the prompt is provided in Appendix B.

For each relevant file, we sent a prompt to the LLaMA-3.3-70B-Instruct model for instruction generation using the method described in Section 4. As a result, we generated 24,854 instructions, which constitutes more than half of the RAG-Instruct dataset Liu et al. (2024).

We observed that some files resulted in only a small number of generated instructions despite their confirmed relevance through manual review. After analyzing the distribution of instruction counts per file, we decided to regenerate instructions for files that initially produced ≤ 3 instructions.

For the second pass of generation, we modified the seed value and lowered the `temperature` (from **1.0** to **0.7**) and `top-p` (from **1.0** to **0.8**) parameters when calling the LLM. The lower values of `temperature` and `top-p` leading to more deterministic answers of the LLM Lee (2025) and

expected to surpass such variance answers as empty output or small amount (less than three) of instruction pairs.

This process generated **806** additional instructions from **91** underperforming files, bringing the total dataset size to **25,633** instructions. This volume is sufficient for further fine-tuning of the generative LLM. The source code of the instructions generation project is available at: `https://github.com/EliseevVadim/InstructionsGenerator`

## 6 ADDING THE CONTEXT TO GENERATED INSTRUCTIONS

In this section, we describe the process of adding relevant context to 25,633 previously generated instructional pairs. First, we built the vector database from the relevant text files. Then, using this database, we found the semantically closest text fragments for each input from the instructions and added them as the *context* field to the dataset.

### 6.1 BUILDING THE VECTOR DATABASE

Before building the vector database we had to choose the retriever model. The "ideal" retriever must have an encoder-only architecture Karpukhin et al. (2020) because we need to obtain qualitative embeddings from it, not to generate text output which is the responsibility of a decoder. Also, a good retriever must be able to process long sequences Khattab & Zaharia (2020) and build embeddings of high dimension $d$ Kaplan et al. (2020).

As we are working with texts written in Russian, it is preferable to use a model, that was trained on Russian texts and not adapted from other languages. Thus, we chose the `Giga-Embeddings-instruct`[7] model as the retriever.

This model is capable of processing contexts up to 4,096 tokens and builds embeddings with a dimensionality of $d = 2,048$. The model is based on GigaChat-Pretrain-3B, where decoder attention has been replaced with encoder attention, and Latent Attention Pooling Lee et al. (2024) has been used for aggregation. This model is also a good choice for the task at hand, as it has only 2.5B parameters, making it easy to run and use on a personal device. Despite its size, it demonstrates high-quality vectorization, ranking second in the ruMTEB[8] benchmark as of December 27, 2024.

Next, we had to choose the chunk size and chunking strategy. Since the vector database created at this stage will be used only for constructing the fine-tuning dataset, we decided to split the documents into relatively small chunks of **500** tokens with an overlap of **50** to avoid overloading the model's context window during fine-tuning. For determining the chunk size, we used the tokenizer of the LLaMA-3.1-8B[9] model, as this is the model planned for fine-tuning on the built dataset.

Additionally, before creating the vector database, it is necessary to ensure that there are no duplicate text files among those used. Otherwise, we risk obtaining identical fragments as top-n documents during vector search, which is detrimental for a RAG system.

After removing duplicate files based on their content and splitting them into chunks, we obtained **9,935** chunks. Using these chunks, we built a vector database with FAISS Douze et al. (2024). Additionally, during the embedding generation process, we applied normalization Yi et al. (2021), which allows us to search for similar chunks using cosine similarity Salton et al. (1975).

### 6.2 ADDING THE CONTEXT TO INSTRUCTIONAL PAIRS

After building the vector database, we expanded our instructional dataset, which contains input-output pairs, by adding relevant context. This context was retrieved by the retriever model after performing similarity search for the input query in the vector database. Thus, during fine-tuning, we adapt the model to the specific retriever in use and train it to extract the correct answer from the retrieved documents. We provide the model with **three** documents along with the query, as we

---

[7]https://huggingface.co/ai-sage/Giga-Embeddings-instruct
[8]https://huggingface.co/spaces/mteb/leaderboard
[9]https://huggingface.co/meta-llama/Llama-3.1-8B-Instruct

do not want to overload the generator's context window. Using a larger number of documents may degrade the final answer quality, because of the potential data noise Izacard et al. (2021).

For each input $q$ from our synthetic instruction dataset we retrieved three semantically closest passages $p$ using cosine similarity $\cos(\theta)$. It can be found as:

$$\cos(\theta) = \frac{p \cdot q}{\|p\|\|q\|} \tag{1}$$

where $p * q$ is a scalar product of vectors that can be found as:

$$\mathbf{p} \cdot \mathbf{q} = \sum_{i=1}^{d} p_i q_i \tag{2}$$

and $\|p\|$, $\|q\|$ are Euclidian norms of vectors $p$ and $q$ that can be found as:

$$\|\mathbf{p}\| = \sqrt{\sum_{i=1}^{d} p_i^2}, \quad \|\mathbf{q}\| = \sqrt{\sum_{i=1}^{d} q_i^2} \tag{3}$$

The cosine similarity value lies within the range of -1 to 1. A value of 1 indicates that the vectors are identical, 0 means they are orthogonal and have no semantic similarity, and -1 signifies that the vectors are opposite. Thus, the higher the value of $\cos(\theta)$, the closer the query $q$ to passage $p$. In our case, for each input, we retrieved the three passages with the highest similarity scores. Then we added these passages to our dataset as the *context* field. The source code of extending instructional pairs with relevant context from the vector database is available at: `https://github.com/EliseevVadim/InstructionsGenerator`

## 7 FUTURE WORK

After successfully building the RAG-powered instruction dataset, we plan to fine-tune the generator model on it using the LoRA method Hu et al. (2021). The chosen model for training is LLaMA-3.1-8B, an open-source model with a relatively small number of parameters, making it feasible to fine-tune even with limited computational resources.

We also plan to build a new vector knowledge database that will include both the relevant text files obtained using the method described in this paper and the textual content of various file formats collected through website crawling.

Since we plan to store large volumes of vectorized textual data and require the ability to regularly update the vector database, we intend to use a persistent storage solution for the vector database. One possible option is pgvector[10].

Using the proposed method of building RAG-powered instruction dataset is expected to enhance the performance of the RAG generator model trained on it while maintaining high final answer quality for the developed intelligent assistant regardless of changes in domain knowledge, as the model will be able to effectively extract information from an easily updatable knowledge base.

## 8 CONCLUSION

In this paper, we presented a method of building a RAG-powered instruction dataset for fine-tuning a generator LLM within the RAG system of an university's intelligent assistant. The novelty of our study primarily lies in the positioning of the developing RAG system, as we aim to train it to work exclusively with domain-specific data about the DSU. While in the related works the existing datasets were predominantly used, we built our own from scratch, using the university's website as a data source and applying a series of transformations described in the paper. We covered the

---

[10]https://github.com/pgvector/pgvector

steps from parsing raw text data from the university's website to extending the classical instruction dataset with relevant context from a vector database that is semantically similar to the input. Using this method, we built a dataset containing over **25,000** input-context-output records, which is ready for use in fine-tuning the generator model. In future work, we plan to fine-tune LLaMA-3.1-8B on this dataset, build a vector database using all the collected data from the university's website, and set up a RAG pipeline that will generate high-quality responses regardless of changes in the knowledge base content.

ACKNOWLEDGMENTS

The study was carried out with the financial support of the Ministry of Education and Science of the Russian Federation within the framework of the state task on the topic "Development and improvement of intelligent classification and forecasting methods for pattern recognition and modeling of information processes" FREM-2024-0001 (Registration number 1023111000141-9-1.2.1)

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

## A    APPENDIX A

For estimation of texts and selecting the relevant ones for instructions generating and building the vector database we used the following prompt. Here is the translated version of the prompt in English, however in our pipeline we used the prompt in Russian. The English version of the prompt:

```
Read the text below, obtained from parsing a web page of an
educational organization, and evaluate its educational value and
usefulness for training an LLM in building an instruction dataset
(i.e., assess how well the provided text can be used to generate
question-answer pairs) that will be used in an intelligent
assistant.

Consider the following criteria in your evaluation:

Content of the text. Pay special attention to links and mentions
of resources from 2022 and later, as they are the most relevant.
Some texts are merely textual representations of links, and
sometimes entire pages contain only links. Such files cannot
be used to build a relevant instruction dataset, so these cases
should receive low scores.

File name. The file name may indicate the topic or importance
of the text and complement its content, especially if it contains
contact details. If the name expands the context of the provided
information to a level where meaningful instructions and answers
can be generated, it increases the text's value.

Rating Scale:

1: The text has no meaningful content, consists only of links, or
serves as a placeholder without useful information.

2: The text contains links and headings with minimal relevant
information, including outdated or irrelevant data.

3: The text is suitable for creating a limited number (fewer than
5) of general instructions but has low value.

4: The text is suitable for creating a significant number of
instructions that reflect both general and specific aspects of
the organization's operations, with a sufficient share of relevant
data.

5: The text is highly structured, contains numerous relevant
details (from 2022 and later), and allows for the creation of a
wide variety of valuable instructions.

Important:

The response must be a single number (from 1 to 5). Comments,
explanations, or additional information are strictly prohibited.

File name:
```

```
***
```

```
Content:
```

```
***
```

## B  APPENDIX B

For generating instructions from the contents of relevant text files we used the following prompt.
Here is the translated version of the prompt in English, however in our pipeline we used the prompt
in Russian. The English version of the prompt:

```
Based on the HTML page data obtained from parsing the website of
Donetsk State University:
```

```
Page Title:
```

```
***
```

```
Date of Last Content Update:
```

```
***
```

```
Content:
```

```
***
```

```
Generate the maximum possible number of instructions with model
responses (including both long and short responses) that fully
cover the provided text, especially its factual aspects.  The
instructions should be suitable for fine-tuning an LLM as an
intelligent assistant for Donetsk State University.  Each
instruction must be formatted as JSON with input (question)
and output (answer).  Avoid generating irrelevant content that
is not specifically related to the activities of Donetsk State
University.  If the text contains no meaningful information (e.g.,
a list of links, advertisements, or other utility data), generate
a stub with "error":  "This text does not contain useful content
for generating instructions".
```

```
IMPORTANT!
```

```
The provided content often pertains to specific local aspects of
the university's activities.  Therefore, DO NOT generate overly
generic questions such as "What programs are offered at DSU?"
or "Who is a professor at DSU?" because the objects described in
specific files are not the only ones in the university's context,
and such questions are counterproductive.
```

```
If the content mentions a faculty member, instead of a question
like "Who is a professor of the Department of Physical Education
and Sports on the <title> page?" with the answer "<Name of the
professor>", generate "Who is <Name of the professor>?" with
the response based on the page content.  Do not respond with "A
professor described on the <title> page"; instead, use the actual
content of the page.  This logic applies to all other instructions
as well.
```

```
Additionally, the page title is provided, which can better reveal
the text's content.  Use it and its connection to the text to
construct higher-quality instructions.
```

```
Moreover, the date of the last content update for each page is
indicated.  Use this information to improve the quality of the
generated instructions.  Avoid asking questions like "When was the
```

page content last updated?" as this question is not relevant; the date should only be used to add context to the content.

Avoid using pronouns or generic nouns when generating questions. Instead, use named entities from the page title or as appropriate to the content. Additionally, when listing facts in the model's response, include all available facts without truncating them with phrases like "and others."

The model being trained will serve as an intelligent assistant for an educational organization. Therefore, this aspect must be the primary consideration when creating instructions and responses|all must be in the context of the specific educational organization being referenced.

The text was obtained by automatically crawling the university website. This prompt is also generated automatically by substituting data obtained during the crawl. Therefore, it may contain irrelevant or useless information. For such texts, generate JSON with the error field containing the value "This text does not contain useful content for generating instructions." Do not generate instructions for meaningless content!

Possible Types of Instructions:

1. Questions to extract facts or details from the text (if factual material is present, questions must cover all of it).
2. Questions requiring analysis or comparison of information.
3. Any other questions applicable for fine-tuning an intelligent assistant for DSU. Instructions and responses must include both concise and detailed formats, fully covering the entire text (every part of the text MUST strictly participate as part of a response).

Output is allowed ONLY in JSON format; any other textual content is strictly prohibited! All output must be presented in Russian language only!

