# OpenReview forum: "Method of Building RAG-Powered Instruction Dataset from Raw Corporate Text Data for LLM Fine-Tuning"
_mathai.club/MathAI/2025/Conference — MathAI 2025 Oral_

### Official Review · Reviewer_ZCMb · 2025-02-26
**Method of Building RAG-Powered Instruction Dataset without any metrics(Updated)**

**Rating:** 6
**Confidence:** 4

**Review:**

The article presents an approach to constructing a training dataset with contextual information for the joint application of RAG and fine-tuning methods.

**Strengths:**
- A detailed and comprehensive algorithm for creating such a dataset is provided, accompanied by an example.
- Each step of the dataset construction process, where necessary, is supplemented with links to open-source code.
- All factual claims are supported with references to the relevant literature throughout the article.
- Examples of prompts are provided where necessary.

**Weaknesses:**
- The article is supposed to be anonymous; however, the author is clearly identified through GitHub links within the article.
- The article concludes without any justification for the effectiveness of the dataset construction algorithm. All such evaluations are deferred to future work.

---

### Official Review · Reviewer_a2NA · 2025-02-26
**Promising approach lacking benchmarking**

**Rating:** 6
**Confidence:** 5

**Review:**

The paper proposes a method to construct a dataset for LLM fine-tuning on arbitrary data for RAG applications. The authors suggest combining a RAG engine with fine-tuning to incorporate domain knowledge into case-specific LLM applications.

**Strengths**
- The paper addresses a relevant problem and proposes a resource-efficient method to tackle it.
- Open-source code is provided along with all prompts, enabling result reproducibility.
- The data generation process is fully described.
- Data cleaning is performed effectively using separate LLM requests.

**Downsides**
- The paper is not fully anonymized; it contains links to personal GitHub accounts and information on the funding source.
- No comparison with other RAG fine-tuning approaches, such as RADIT (https://arxiv.org/abs/2310.01352), which also proposes tuning LLMs on query-context-answer sequences and includes retriever tuning. Additionally, long-context models should be considered, as linear attention mechanisms and alike enable entire database to be incorporated in context.
- While data preprocessing is well executed, file chunking is not utilized during dataset preparation, unlike in the RAG database-building stage. Large text segments are removed, though they may contain crucial information, potentially limiting the method’s generalizability.
- The effect of temperature adjustments on results is not examined. Including a human-assessed data sample would improve reliability.
- The prompt lacks labeled examples, reducing stability and making assessments less reliable. A better approach might involve first extracting all factual data before generating context-based questions.

**Conclusions**
The paper is well-structured and provides a detailed explanation of the data generation process, ensuring reproducibility for creating a RAG fine-tuning dataset. The proposed method is computationally efficient, enhancing accessibility. However, an experimental assessment of the dataset and fine-tuning results would strengthen the study. Additionally, the novelty of the method could be expressed more explicitly, as existing RAG fine-tuning techniques are not discussed.

**Remarks**
- Prompts could be included as text so they can be translated into English. Moving prompt figures to an attachment would also improve clarity.
- Line 062: "However, both of them suffering from ..." — a verb is missing.
- Line 095: It would be better to separate the retriever from the encoder. The retriever operates on a prepared vector database and does not need to vectorize documents, whereas the encoder remains necessary for query vectorization (as stated in Line 098). The retriever typically uses ANN or similar methods to retrieve similar documents.
- Line 110: "The best way to deal with such situations for the model to acknowledge that it lacks the information needed to generate the answer" → "The best way to deal with such situations is for the model to acknowledge that it lacks the information needed to generate the answer."
- Line 124: "The best fine-tuning approach" — It would be helpful to specify the criteria for determining the optimal approach rather than using the vague term "best approach."
- Consider whether a generative approach is suitable for determining if a page contains relevant information. For a purely discriminative task, fine-tuning a text encoder on sample data could be a low-resource alternative to API requests.

UPD from 07.03

Changed the rating according to the authors' comments

As the open-source code is a necessary addition to the paper, for the next submissions authors are advised to use repo anonimization services (e.g., https://anonymous.4open.science/)

---

### Official Review · Reviewer_pxHU · 2025-02-27
**A method for generating AI training data from university websites has been proposed, but there is a need for more testing, originality, and a broader focus**

**Rating:** 6
**Confidence:** 4

**Review:**

### 1. **Summary**
The paper proposes a method to create a RAG-powered instruction dataset for fine-tuning LLMs in corporate QA systems. By combining retrieval-augmented generation (RAG) with supervised fine-tuning, the authors generate synthetic input-context-output triples from university website data. The five-step pipeline includes data crawling, relevance filtering, instruction generation, vector database construction, and context augmentation. The resulting dataset contains 25,633 entries tailored for training a generator LLM to better utilize retrieved context, addressing limitations of standalone RAG/fine-tuning approaches.

---

### 2. **Strengths and Weaknesses**
**a. Originality**
- **Strength**: Combines RAG context retrieval with instruction tuning in a novel way to Russian-language corporate data.
- **Weakness**: The core idea (RAG-augmented fine-tuning) builds on established concepts (e.g., RAFT https://arxiv.org/abs/2403.10131), with limited discussion of how this differs from similar hybrid approaches.

**b. Quality**
- **Strength**: Technically sound pipeline with open-sourced code. Relevance scoring via LLM evaluation and correlation analysis adds rigor.
- **Weakness**: No end-to-end evaluation of the fine-tuned model’s performance in a RAG system. The claim that the method "will ensure more accurate performance" remains unvalidated.

**c. Clarity**
- **Strength**: Well-structured methodology with clear steps. Figures (e.g., distribution of file evaluations) aid understanding.
- **Weakness**: Section 3 (Method) lacks a schematic overview of the pipeline.

**d. Significance**
- **Strength**: Addresses a practical challenge (domain-specific QA systems) with a scalable approach. The dataset and codebase enable reproducibility.
- **Weakness**: Impact is limited to a single domain (university data); broader applicability to other corporate settings is underexplored.

---

### 3. **Questions for Authors**
1. How does the model handle queries requiring context *not* present in the training data or vector DB?
2. Why was LLaMA-3.3-70B chosen for instruction generation over smaller models (e.g., LLaMA-3.1-8B)?
3. What is the impact of chunk size (500 tokens) on retrieval quality during fine-tuning?
4. Could the synthetic instruction generation introduce biases (e.g., overfitting to website content)?

---

### 4. **Limitations**
- **Addressed**: Acknowledges the lack of fine-tuning results and plans future work.
- **Missing**:
  - No discussion of potential biases in synthetic data generation.
  - Scalability to larger corpora (14.3GB data mentioned but only 2,912 texts used).
- **Suggestion**: Include a bias analysis and test the pipeline on a public benchmark (e.g., nvidia/ChatRAG-Bench and/or its analogues in Russian).

---

### 5. **Ethical Concerns**

No major issues flagged. However, the use of university data (even public) should include a privacy statement.

---

### 6. **Soundness**

Methodologically solid but lacks end-to-end evaluation.

---

### 7. **Presentation**

The materials are presented well, but links to GitHub repositories may compromise anonymity.

---

### 8. **Contribution**

Practical solution with open resources, but incremental over prior RAG/fine-tuning work.

---

### 9. **Overall Score**

The paper presents a useful method for RAG-augmented dataset creation with open-source tools. While the technical execution is competent, the lack of empirical validation and minor presentation issues limit its impact. A rebuttal addressing evaluation plans and limitations could strengthen the case for acceptance.

---

---

### Author Rebuttal · Authors · 2025-03-05

After carefully reviewing the feedback on our paper, we have made several updates to its content. In this response, we will outline the most significant changes and clarify key aspects of our research that may have been unclear.

As many reviewers suggested, including English translations of the prompts would improve the clarity of our study. In response to this recommendation, we have added English translations for both tasks — selecting relevant texts and generating instructions based on those texts. These translated prompts can be found in Appendix A and Appendix B, respectively.

Additionally, some reviewers raised questions and provided suggestions regarding the evaluation of the proposed method, particularly concerning the performance assessment of the fine-tuned model within the RAG system. However, as stated in the paper, our study focuses specifically on the process of constructing a domain-specific dataset suitable for tuning a RAG generator from scratch, rather than on the tuning process itself. The tuning procedure and its outcomes will be addressed in future work.

Several reviewers also pointed out that the novelty of our study was not articulated as clearly as necessary. To address this, we have revised the conclusion to provide a more explicit discussion of our contributions and the positioning of our results. Furthermore, we have added additional comparisons with existing studies that focus on dataset creation for training generator models or both retrievers and generators in RAG systems.

The revised version of the paper, incorporating these updates as well as minor refinements, has been submitted for further review.

---

### Decision · Program_Chairs · 2025-03-08

**Decision:**

Accept (Oral)

**Comment:**

Your article has been accepted and you can make a presentation on the article. All articles will be sorted by rating and within the available conference places one author from each article will be invited. If there are not enough places, then you will either have the opportunity to present remotely or come at your own expense!